# The Differences in Physical Activity Preferences and Practices among High versus Low Active Adolescents in Secondary Schools

Karel Frömel [1,2], Dorota Groffik [2], Michal Kudláček [1], Michal Šafář [1], Anna Zwierzchowska [2] and Josef Mitáš [1,*]

1    Faculty of Physical Culture, Palacký University Olomouc, tř. Míru 117, 77111 Olomouc, Czech Republic; karel.fromel@upol.cz (K.F.); michal.kudlacek@upol.cz (M.K.); michal.safar@upol.cz (M.Š.)
2    Institute of Sport Sciences, The Jerzy Kukuczka Academy of Physical Education, Mikołowska 72a, 40-065 Katowice, Poland; d.groffik@awf.katowice.pl (D.G.); a.zwierzchowska@awf.katowice.pl (A.Z.)
*    Correspondence: josef.mitas@upol.cz; Tel.: +420-58563-6745

**Abstract:** This study aimed to (a) identify the differences in the preferred, practiced, and organized physical activity (PA) between high and low physically active Czech and Polish boys and girls and to (b) identify which types of PA are likely to be recommended by low active boys and girls. The research was carried out between 2010 and 2019 in the Czech Republic and Poland and included 6619 participants aged 15 to 19 years. The preferences and practices of the different types of PA and participation in organized PA were identified using a PA preference questionnaire and weekly PA was identified using the International Physical Activity Questionnaire-long form. Czech and Polish boys and girls who were low active preferred less physically demanding and more health-oriented PA compared with high active individuals. Power exercises and participation in organized PA are the most significant predictors of achieving at least $3 \times 20$ min of vigorous PA per week among low active individuals in both countries. Monitoring the trends in the preferred and practiced types of PA among boys and girls is crucial for the effective promotion of PA to low active boys and girls and positive changes in physical education and school health policy.

**Keywords:** preferences; physically inactive adolescents; types of physical activity; gender differences; school health policy; sustainable health

## 1. Introduction

Chronic insufficient physical activity (PA) among adolescents has been observed across continents [1–3]. Insufficient PA among adolescents has been documented in summary reviews of the European Union countries [4,5] and regional studies in Visegrád countries, including the Czech Republic, Hungary, Poland, Slovakia [6], Germany [7], Sweden [8], and Slovenia [9], as well as in other countries.

Naturally, decreasing PA in adolescents is associated with the achievement of PA recommendations. According to Guthold et al. [2], globally, the most frequent recommendation of 60 min per day of moderate-to-vigorous PA (MVPA) is not achieved in 78% of boys and 85% of girls. In the European Union, the MVPA recommendation is not achieved by 82% of boys and 89% of girls [10]. Most European countries support this PA recommendation or even longer durations of PA per day and the inclusion of at least 20 min of vigorous PA 3 times per week for strengthening muscles and bones [11]. When assessing meeting of PA recommendation, it is also necessary to consider age, gender, region, and specific target population groups [12]. PA recommendations should also be placed in context with other indicators of a healthy adolescent lifestyle. For example, the recommendation of ≥60 min of PA every day, daily consumption of fruit and vegetables, spending <2 h daily immersed in screen-based behaviors, and abstinence from alcohol and tobacco products are a set of indicators of a healthy lifestyle that is achieved by only 1.9% of European adolescents [13]. Although this multi-criteria recommendation for a healthy adolescent lifestyle is simply

understandable, achieving all criteria is difficult. This emphasizes the need for a cautious approach to meeting PA recommendations, especially in low active (LA) adolescents.

A research area that is still neglected is the associations between PA achievement by adolescents and types of PA. There is a lack of studies that analyzed associations between the level of PA and the preferred and practiced types of PA that are most favored among LA adolescents. This issue has been addressed in a research study aimed at the associations between PA level and participation in organized PA (OPA) [14]. The analyses of the positives and negatives of different types of PA within OPA among adolescents with low physical activities have been of great relevance, especially during and after the pandemic. The pandemic has resulted in a further decrease in adolescents' PA [15–18], with a negative impact on adolescent mental health [19–21]. After the pandemic or in the case of a new pandemic, the return to a physically active lifestyle, especially for LA adolescents, in different regions and under varying educational conditions will be extremely difficult.

Secondary schools will play a crucial role in eliminating the negative consequences of the pandemic on LA adolescents and in the health promotion of every student [22]. The restoration or creation of regular PA habits in LA adolescents cannot be left to families and leisure centers. Maintaining PA habits is dependent on the level of school physical education (PE), the effectiveness of comprehensive school PA programs, extracurricular school PA programs, and other school-based activities.

This study aimed to (a) identify the differences in preferred, practiced, and organized PA between high and low physically active Czech and Polish boys and girls and (b) to identify which types of PA are likely to be recommended by low active Czech and Polish boys and girls.

## 2. Materials and Methods

### 2.1. Participants and Setting

The research was conducted in 94 secondary schools in the Czech Republic and 83 secondary schools in Poland between 2010 and 2019. Each year, seven to ten secondary schools on average from both countries participated. Secondary schools were selected on the basis of long-term cooperation with university departments concerning the organization of their students' teaching practices. The research involved a total of 4258 Czech and 2361 Polish adolescents. Body mass index (BMI) was calculated using the WHO BMI z-scores for adolescents. A total of 20.4% of boys and 11.2% of girls suffered from overweight/obesity. Each year, the research involved 571–859 adolescents. The number of participants involved in autumn (September–November) and spring (March–May) was similar. School administrations, parents, and participants confirmed their agreement to participate in the research by providing written informed consent. As the research study was presented as education and a source of important information for the school administrations, the research study included all students in the selected groups who were present on the day of the research.

The research was conducted under similar conditions at both Czech and Polish secondary schools at the time of the habitual weather (same season) and habitual educational weekly program. An identical pair of researchers led the research study at each school. The participants were informed about the aims and benefits of the research during an introductory meeting in the computer room. The participants were also informed about the presentation of the results after the completion of the research and how the results would be used to improve school PE and school lifestyle. During the introductory meeting, the participants were registered in the 'International Database for Research and Educational Support' web application (Indares) (www.indares.com).

### 2.2. Questionnaire Measures

The "International Physical Activity Questionnaire-long form" (IPAQ-LF) for adolescents was used [23,24]. Both Czech and Polish versions of the IPAQ-LF were processed in compliance with applicable translation requirements [25] and empirically verified in

international comparative studies [14,26]. The coefficients of concurrent validity between overall weekly PA (METs-min) using the IPAQ-LF questionnaire and weekly step count (steps/week) in both versions of the questionnaire were based on Pearson's correlation coefficient in the range of r = 0.231–0.283. Cronbach's alpha for internal consistency reliability was α = 0.848 for the Polish version and α = 0.845 for the Czech version. The results of the IPAQ-LF questionnaire were processed as per the manual but with the following adjustments: The MET-min of vigorous PA (VPA) was multiplied by six; the maximum MET-min per week was limited to 20,000 MET-min; and the maximum average daily sum of PA, transportation, sitting, and passive commuting was set at 960 min/day. A total of 490 respondents who did not meet the predetermined criteria were excluded.

The participants were divided into three tertile-based groups (low, moderate, and high active) based on their total weekly PA level and separately for girls and boys. We used "low PA" to describe the less active adolescents because it is more apt than the frequently used term "physically inactive" adolescents [27]. The differences between groups of LA and high active (HA) adolescents were reported only for the most frequent types of PA and for types of PA with the largest differences between the two groups.

Weekly PA recommendations were modified according to Healthy People 2020 [28], Physical Activity Guidelines for Americans [29], and in compliance with the recommendations for the different PA types [30]. For the LA adolescents, the most relevant and stringent recommendation on weekly vigorous PA was determined as at least three or more days of at least 20 min of vigorous PA (VPA) per week (3 × 20 min VPA). Another reason for the inclusion of VPA recommendations was to respect the associations between physical fitness, PA, and mental health of adolescents [31]. The IPAQ-LF questionnaire was completed by all respondents.

In order to identify preferences for the types of PA, a questionnaire on preference for physical activity (QPAP) was used. Both Czech and Polish versions of the questionnaire were standardized in the respective country for youth aged 12 years and older [32,33]. The highest stability between the first and the second measurements were in the group of team sports ($r_s$ = 0.76–0.81), and the lowest stability was then recorded for the group of rhythmic and dancing activities ($r_s$ = 0.62–0.68). The questionnaire instructs respondents to choose their five most preferred types of PA in the following categories: individual PA; team PA; fitness PA; PA in water; PA in nature; martial PA; rhythm and dance PA; and total PA. Due to the different number of PA types in each category, the preferences of PA types cannot be compared between categories but only within a single category. The questionnaire included 90 types of PA. Respondents can assign any other PA to the most appropriate type of PA from the list. In this study, only PA types that were ranked as the most preferred (ranked first) were considered. The following questions from the QPAP questionnaire were also used: "Indicate your participation in regular and organized PA (under supervision of a teacher, trainer, or coach) during the week in your free time during the past 12 months, except for holidays" and "Indicate the most frequent non-organized PA pursued in your free time during the past 12 months (specify the types of PA in summer and winter)." Regular organized PA included any PA in an organized form except for PE lessons. In this study, we present selected types of PA according to the magnitude of differences and the number of participants.

### 2.3. Data Analysis

Statistica version 13 (StatSoft, Prague, Czech Republic) and SPSS version 25 (IBM Corp., Armonk, NY, USA) were used for statistical analysis. We used descriptive characteristics for the preferred and practiced types of PA. The Kruskal–Wallis ANOVA was used to determine the structure of weekly PA. The differences between the groups of participants with different PA levels and their PA preferences were determined using cross tables. The likelihood of achieving VPA recommendations was assessed via binary logistic regression analyses using the forward stepwise (likelihood ratio) method (due to the high number of categorical covariates and the lower tendency for errors). The $\eta^2$ effect size coefficients

were evaluated as follows: $0.01 \leq \eta^2 < 0.06$ small effect size; $0.06 \leq \eta^2 < 0.14$ medium effect size; and $\eta^2 \geq 0.14$ large effect size. The sample size met the requirements for the application of binary logistic regression [34]. The level of significance was set at $p \leq 0.05$. Logically significant differences were >5% in the preferred or practiced PA.

### 3. Results

*3.1. Characteristics of Low and High Physically Active Boys and Girls*

The characteristics of low and high physically active boys and girls are based on the characteristics of the basic sample (Table 1).

**Table 1.** Sample characteristics.

| Gender | Country | *n* | Age (years) | | Weight (kg) | | Height (cm) | | PA (MET-min/day) | | Sitting (min/day) | |
|---|---|---|---|---|---|---|---|---|---|---|---|---|
| | | | **M** | **SD** | **M** | **SD** | **M** | **SD** | **M** | **SD** | **M** | **SD** |
| Boys | Czech Republic | 1614 | 16.7 | 1.3 | 69.8 | 12.6 | 178.1 | 8.5 | 864 | 664 | 386 | 127 |
| | Poland | 1072 | 16.3 | 0.8 | 67.6 | 12.5 | 176.9 | 7.5 | 1010 | 746 | 358 | 151 |
| Girls | Czech Republic | 2644 | 16.8 | 1.2 | 58.8 | 9.3 | 167.0 | 6.7 | 729 | 579 | 392 | 121 |
| | Poland | 1289 | 16.3 | 0.7 | 57.1 | 8.9 | 165.7 | 6.2 | 878 | 693 | 368 | 144 |

M: mean; SD: standard deviation; PA: physical activity.

The differences in BMI between LA and HA boys ($p = 0.215$) and between LA and HA girls ($p = 0.081$) were not statistically significant. According to self-reported measurements, 20.4% of boys and 11.2% of girls were overweight or obese. Surprisingly, 7.6% of LA boys as opposed to 13.1% of HA boys admitted smoking ($\chi^2 = 14.53$, $p < 0.001$, $\eta^2 = 0.008$). Similarly, 7.0% of LA girls as opposed to 11.3% of HA girls reported smoking ($\chi^2 = 14.48$, $p < 0.001$, $\eta^2 = 0.006$).

*3.2. Differences in the Preference of PA Types*

The greatest difference in the preference of PA types between LA and HA boys was observed in PA types that were less demanding in terms of movement and fitness (Table 2). Czech and Polish LA boys equally preferred badminton, volleyball, bowling, downhill skiing, fitness walking, hiking, health exercise, and karate. In contrast, Czech and Polish HA boys preferred track and field activities, cycling, snowboarding, soccer, bodybuilding PA, power exercises, boxing, and kickboxing. Statistically significant differences were evident in the preference for soccer. A logically significant difference was observed in the preference for running/jogging (40.0% of HA boys; 34.2% of LA boys).

The differences in the preference of PA types between LA and HA girls were less significant compared to boys (Table 3). Czech and Polish LA girls jointly preferred badminton, downhill skiing, floorball, yoga, health swimming, karate, Latin American dance, and standard dance. On the contrary, Czech and Polish HA girls preferred track and field, artistic gymnastics, basketball, volleyball, power exercise, and modern dance.

In the summary categories for PA types, Czech HA boys preferred team PA, while Polish HA boys preferred martial arts (Figure 1A). The results also suggest that LA boys were more likely to prefer individual PA and PA in nature and water. For girls, in the summary categories for PA types, the only statistically significant difference was observed in the preference of fitness activities by Czech HA girls as opposed to Czech LA girls (Figure 1B).

**Table 2.** Preferences of selected types of PA in Czech and Polish LA boys (tertile 1) and HA boys (tertile 3).

| Type of PA | Physical Activity of Czech Boys | | | | | Physical Activity of Polish Boys | | | | |
|---|---|---|---|---|---|---|---|---|---|---|
| | **Low** | **High** | $\chi^2$ | *p* | $\eta^2$ | **Low** | **High** | $\chi^2$ | *p* | $\eta^2$ |
| | ***n* (%)** | ***n* (%)** | | | | ***n* (%)** | ***n* (%)** | | | |
| Individual PA | | | | | | | | | | |
| Cycling | 71 (13.2) | 92 (17.1) | 3.19 | 0.074 | 0.003 | 46 (12.9) | 49 (13.73) | 0.12 | 0.723 | <0.001 |
| Downhill skiing | 60 (11.2) | 45 (8.4) | 2.37 | 0.123 | 0.002 | 30 (8.4) | 21 (5.88) | 1.68 | 0.195 | 0.002 |
| Track and field | 58 (10.7) | 72 (13.48) | 1.71 | 0.190 | 0.002 | 31 (8.7) | 44 (12.32) | 2.56 | 0.110 | 0.004 |
| Snowboarding | **44 (8.2)** | **67 (12.5)** | **5.31** | **0.021** | 0.005 | 11 (3.1) | 12 (3.36) | 0.05 | 0.827 | <0.001 |
| Badminton | **44 (8.2)** | **12 (2.2)** | **19.29** | **<0.001** | 0.018 * | 15 (4.2) | 10 (2.80) | 1.02 | 0.312 | 0.001 |
| Bowling | **28 (5.2)** | **13 (2.4)** | **5.71** | **0.017** | 0.005 | **24 (6.7)** | **12 (3.36)** | **4.18** | **0.041** | 0.006 |
| Team PA | | | | | | | | | | |
| **Soccer** | **143 (26.6)** | **183 (34.0)** | **7.04** | **0.008** | 0.007 | **142 (39.7)** | **170 (47.6)** | **4.60** | **0.032** | 0.006 |
| Volleyball | 41 (7.6) | 32 (5.9) | 1.19 | 0.275 | 0.001 | 64 (17.88) | 63 (17.65) | 0.01 | 0.936 | <0.001 |
| Fitness and health-oriented PA | | | | | | | | | | |
| Running/Jogging | 184 (34.2) | 215 (40.0) | 3.83 | 0.050 | 0.004 | 89 (24.9) | 87 (24.37) | 0.02 | 0.879 | <0.001 |
| Power exercises | 156 (29.0) | 178 (33.1) | 2.10 | 0.147 | 0.002 | 78 (21.8) | 85 (23.81) | 0.42 | 0.519 | 0.001 |
| **Health swimming** | **123 (22.9)** | **91 (16.9)** | **5.97** | **0.015** | 0.006 | 108 (30.2) | 111 (31.09) | 0.07 | 0.789 | <0.001 |
| **Fitness walking** | **50 (9.3)** | **21 (3.9)** | **12.68** | **<0.001** | 0.019 * | 30 (8.4) | 20 (5.60) | 2.12 | 0.145 | 0.003 |
| Body styling | 37 (6.9) | 46 (8.6) | 1.06 | 0.304 | 0.001 | **37 (10.3)** | **85 (23.81)** | **22.9** | **<0.001** | 0.032 * |
| **Health exercise** | **17 (3.2)** | **4 (0.7)** | **8.21** | **0.004** | 0.008 | **29 (8.1)** | **14 (3.92)** | **5.52** | **0.019** | 0.008 |
| Outdoor activities | | | | | | | | | | |
| **Cycling tourism** | **76 (14.1)** | **102 (19.0)** | **4.55** | **0.033** | 0.004 | 41 (11.5) | 43 (12.04) | 0.06 | 0.806 | <0.001 |
| **Golf** | **34 (6.3)** | **17 (3.2)** | **5.95** | **0.015** | 0.006 | 10 (2.8) | 15 (4.20) | 1.05 | 0.305 | 0.002 |
| **Walking tourism** | **47 (8.7)** | **23 (4.3)** | **8.80** | **0.003** | 0.008 | 27 (7.54) | 20 (5.60) | 1.10 | 0.295 | 0.002 |
| Martial arts | | | | | | | | | | |
| **Boxing** | **87 (16.2)** | **127 (23.6)** | **9.33** | **0.002** | 0.009 | **81 (22.63)** | **107 (29.97)** | **4.98** | **0.026** | 0.007 |
| Kickboxing | 74 (13.8) | 91 (16.9) | 2.09 | 0.150 | 0.002 | **38 (10.61)** | **64 (17.93)** | **7.81** | **0.005** | 0.011 * |
| Karate | 69 (12.8) | 50 (9.3) | 3.41 | 0.648 | 0.003 | 39 (10.89) | 33 (9.24) | 0.54 | 0.463 | 0.001 |

PA: physical activity; $\chi^2$: Pearson's chi-squared test; *p*: significance level; $\eta^2$: effect size; * $0.01 \le \eta^2 < 0.06$ small effect size.

**Table 3.** Preferences of selected types of PA in Czech and Polish LA girls (tertile 1) and HA girls (tertile 3).

| Type of PA | Physical Activity of Czech Girls | | | | | Physical Activity of Polish Girls | | | | |
|---|---|---|---|---|---|---|---|---|---|---|
| | **Low** | **High** | $\chi^2$ | *p* | $\eta^2$ | **Low** | **High** | $\chi^2$ | *p* | $\eta^2$ |
| | ***n* (%)** | ***n* (%)** | | | | ***n* (%)** | ***n* (%)** | | | |
| Individual | | | | | | | | | | |
| Downhill skiing | **126 (14.29)** | **96 (10.90)** | **4.60** | **0.032** | 0.003 | 21 (4.88) | 22 (5.13) | 0.03 | 0.870 | <0.001 |
| Track and field | 92 (10.43) | 100 (11.35) | 0.38 | 0.535 | <0.001 | **38 (8.84)** | **57 (13.29)** | **4.32** | **0.038** | 0.005 |
| Snowboarding | 74 (8.39) | 98 (11.12) | 3.74 | 0.053 | 0.002 | 26 (6.05) | 20 (4.66) | 0.81 | 0.367 | 0.001 |
| Badminton | 62 (7.03) | 52 (5.90) | 0.93 | 0.336 | 0.001 | **41 (9.53)** | **18 (4.20)** | **9.57** | **0.002** | 0.011 * |
| Gymnastics | 35 (3.97) | 51 (5.79) | 3.15 | 0.076 | 0.002 | **39 (9.07)** | **66 (15.38)** | **7.98** | **0.005** | 0.009 |
| Team | | | | | | | | | | |
| Volleyball | 312 (35.37) | 323 (36.66) | 0.32 | 0.570 | <0.001 | 205 (47.67) | 212 (49.42) | 0.26 | 0.609 | <0.001 |
| Basketball | 92 (10.43) | 118 (13.39) | 3.69 | 0.055 | 0.002 | 81 (18.84) | 83 (19.35) | 0.36 | 0.849 | <0.001 |
| Floorball | 87 (9.86) | 80 (9.08) | 0.32 | 0.574 | <0.001 | **16 (3.72)** | **4 (0.93)** | **7.34** | **0.007** | 0.009 |

**Table 3.** *Cont.*

| Type of PA | Physical Activity of Czech Girls | | | | | Physical Activity of Polish Girls | | | | |
|---|---|---|---|---|---|---|---|---|---|---|
| | Low | High | $\chi^2$ | *p* | $\eta^2$ | Low | High | $\chi^2$ | *p* | $\eta^2$ |
| | *n* (%) | *n* (%) | | | | *n* (%) | *n* (%) | | | |
| Fitness and health | | | | | | | | | | |
| Health swimming | 217 (24.60) | 186 (21.11) | 3.05 | 0.081 | 0.002 | 152 (35.35) | 143 (33.33) | 0.39 | 0.534 | <0.001 |
| Power exercises | 159 (18.03) | 184 (20.89) | 2.30 | 0.130 | 0.001 | 52 (12.09) | 59 (13.75) | 0.53 | 0.468 | 0.001 |
| Yoga | 121 (13.72) | 87 (9.88) | 6.26 | 0.123 | 0.004 | 31 (7.21) | 22 (5.13) | 1.61 | 0.205 | 0.002 |
| Spinning | **32 (3.63)** | **50 (5.68)** | **4.17** | **0.041** | 0.003 | 18 (4.19) | 10 (2.33) | 2.34 | 0.126 | 0.003 |
| Outdoor activities | | | | | | | | | | |
| Cycling tourism | **96 (10.88)** | **68 (7.72)** | **5.24** | **0.022** | 0.003 | 28 (6.51) | 36 (8.39) | 1.10 | 0.294 | 0.001 |
| Walking tourism | 63 (7.14) | 54 (6.13) | 0.73 | 0.393 | <0.001 | 42 (9.77) | 46 (10.72) | 0.21 | 0.644 | <0.001 |
| Martial arts | | | | | | | | | | |
| Boxing | 154 (17.46) | 184 (20.89) | 3.34 | 0.068 | 0.002 | **78 (18.14)** | **109 (25.41)** | **6.66** | **0.010** | 0.008 |
| Karate | 149 (16.89) | 130 (14.76) | 1.51 | 0.219 | 0.001 | 73 (16.98) | 58 (13.52) | 1.99 | 0.159 | 0.002 |
| Dance | | | | | | | | | | |
| Modern dance | **236 (26.76)** | **287 (32.58)** | **7.15** | **0.007** | 0.004 | **110 (25.58)** | **161 (37.53)** | **14.20** | **<0.001** | 0.017 * |
| Latin American dance | **180 (20.41)** | **139 (15.78)** | **6.38** | **0.012** | 0.004 | 22 (5.12) | 17 (3.96) | 0.66 | 0.417 | 0.001 |
| Standard dance | **80 (9.07)** | **48 (5.45)** | **8.59** | **0.003** | 0.005 | 37 (8.60) | 33 (7.69) | 0.24 | 0.625 | <0.001 |

PA: physical activity; $\chi^2$: Pearson's chi-squared test; *p*: significance level; $\eta^2$: effect size; * $0.01 \leq \eta^2 < 0.06$ small effect size.

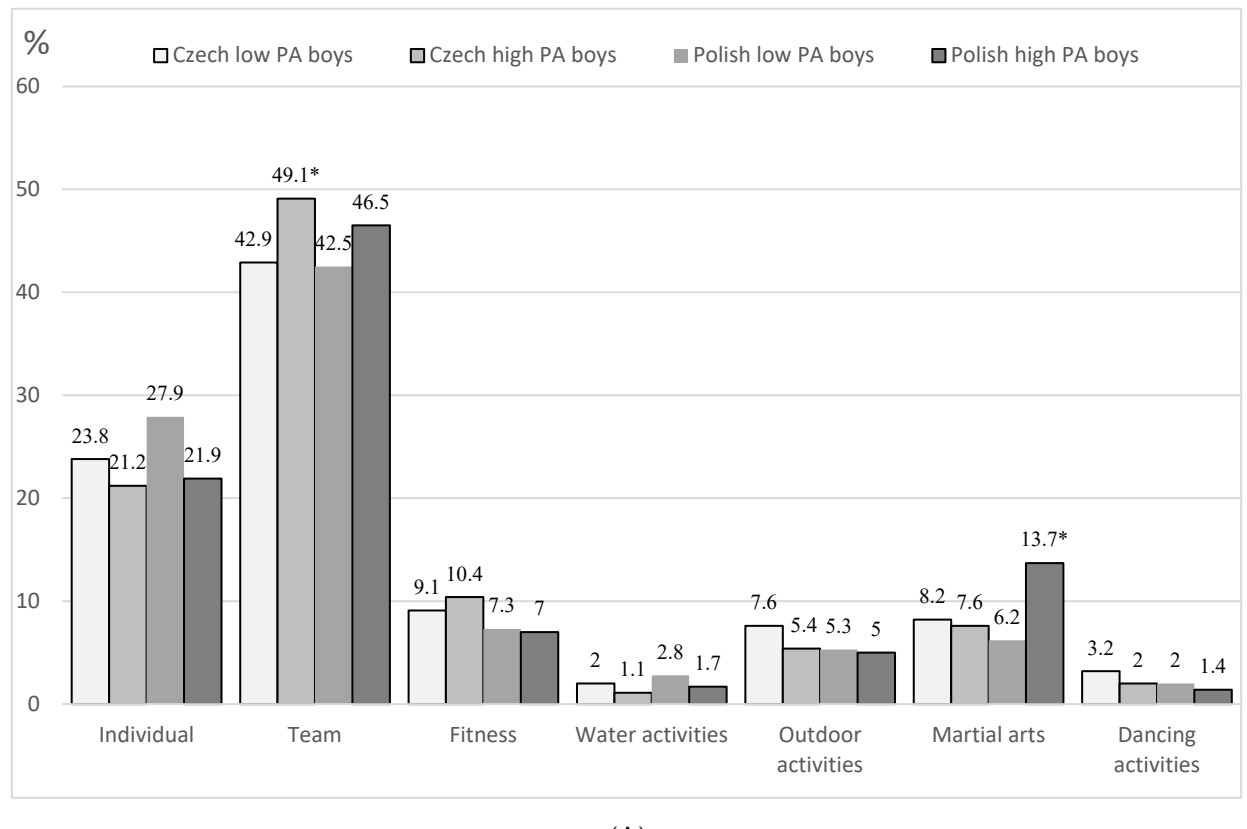

**(A)**

**Figure 1.** *Cont.*

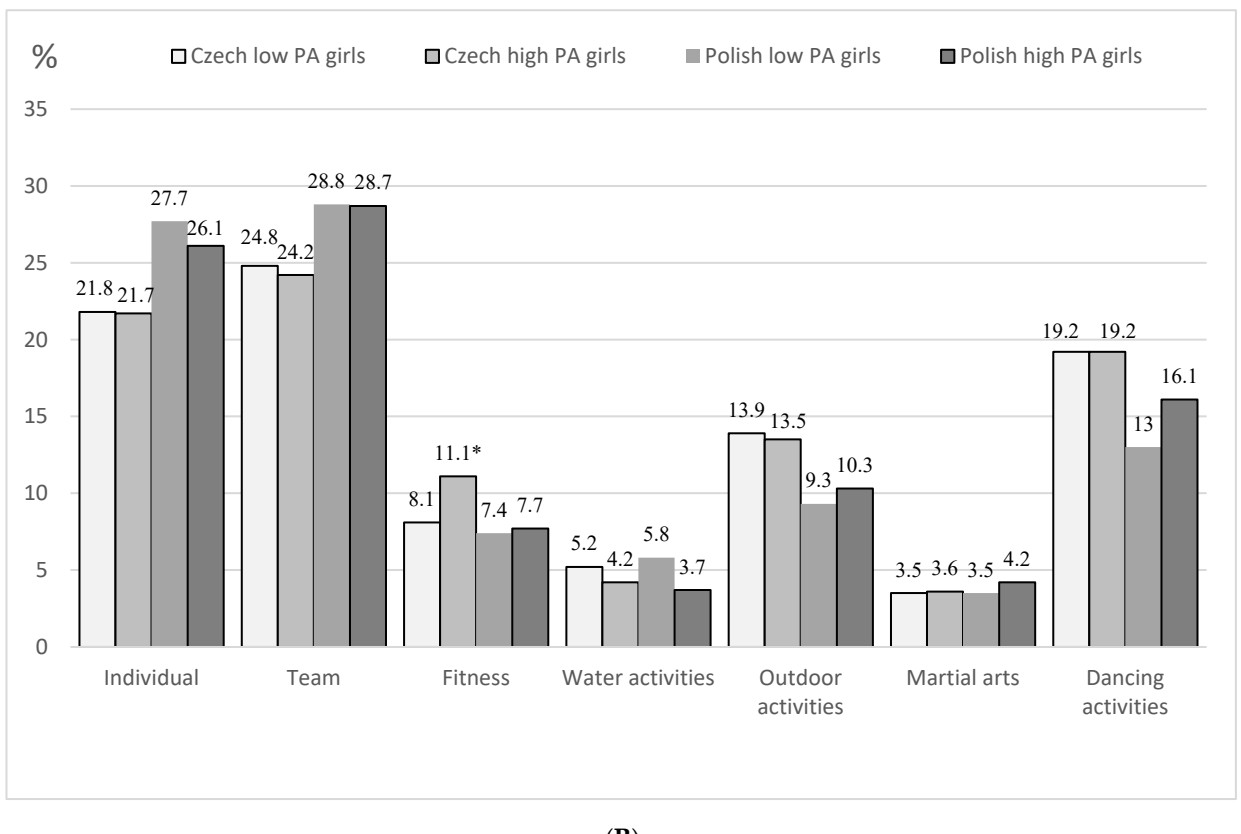

(**B**)

**Figure 1.** Preferences of physical activity (PA) types among Czech and Polish low versus high active boys (**A**) and girls (**B**) by category (* $p \leq 0.05$).

### 3.3. Differences in the Practice of PA Types

Logically significant differences between LA and HA boys in both countries were observed in the practice of PA in winter in power exercises and still in snowboarding among Czech boys (Table 4). The most popular activity among Czech LA boys was cycling in summer and downhill skiing in winter. The most popular activity among Polish LA boys was soccer, both in summer and winter.

Czech girls preferred cycling in summer and downhill skiing in winter, while Polish girls preferred track and field in summer and skating in winter.

### 3.4. Differences in Participation in Organized Physical Activity

A total of 42.9% of Czech LPA boys did not participate in OPA (23.8% HA boys). Better participation in OPA was observed in Poland. A total of 37.7% of LA boys and 17.9% of HA boys did not participate in OPA. Among the boys from both countries, the highest participation rate in OPA was observed for soccer. The greatest participation rate in OPA among Czech girls was observed in dance and among Polish girls in volleyball (Table 5).

### 3.5. Achievement of Vigorous Physical Activity Recommendations

Participation in OPA increased the likelihood of meeting the VPA recommendations in LA Czech and Polish boys and Czech girls (Table 6). A significant predictor of attaining VPA recommendations in Czech LA boys was soccer; in Polish LA boys it was badminton and power exercises; in Czech LA girls it was power exercise; and in Polish LA girls it was cycling.

**Table 4.** The most practiced types of physical activity in summer and winter among LA (tertile 1) and HA (tertile 3) Czech and Polish boys and girls.

| | Summer Physical Activity | | | | Winter Physical Activity | | |
|---|---|---|---|---|---|---|---|
| | Low PA | High PA | | | Low PA | High PA | |
| Type of PA | *n* (%) | *n* (%) | *p* | Type of PA | *n* (%) | *n* (%) | *p* |
| Czech boys | | | | | | | |
| Cycling | 94 (17.5) | 90 (16.7) | 0.886 | Downhill skiing | 102 (19.0) | 95 (17.7) | 0.815 |
| Track and field | 78 (14.5) | 87 (16.2) | 0.763 | Track and field | 68 (12.6) | 59 (11.0) | 0.781 |
| Soccer | 71 (13.2) | 83 (15.4) | 0.698 | Power exercise | **50 (9.3)** | **77 (14.3)** | 0.402 |
| Power exercises | 59 (11.0) | 60 (11.2) | 0.972 | Soccer | 34 (6.3) | 27 (5.0) | 0.828 |
| Walking | 32 (5.9) | 11 (2.0) | 0.605 | Snowboarding | **30 (5.6)** | **58 (10.8)** | 0.420 |
| Swimming | 32 (5.9) | 34 (6.3) | 0.946 | Floorball | 28 (5.2) | 23 (4.3) | 0.881 |
| Floorball | 26 (4.8) | 24 (4.5) | 0.960 | Walking | 27 (5.0) | 3 (0.6) | 0.729 |
| Badminton | 19 (3.5) | 4 (0.7) | 0.766 | Swimming | 25 (4.6) | 18 (3.3) | 0.831 |
| Basketball | 18 (3.3) | 23 (4.3) | 0.869 | Cross-country skiing | 19 (3.5) | 21 (3.9) | 0.947 |
| Tennis | 17 (3.2) | 18 (3.3) | 0.987 | Ice hockey | 17 (3.2) | 26 (4.8) | 0.797 |
| Polish boys | | | | | | | |
| Soccer | 92 (25.7) | 103 (28.9) | 0.617 | Soccer | 50 (14.0) | 54 (15.1) | 0.874 |
| Track and field | 32 (8.9) | 26 (7.3) | 0.825 | Downhill skiing | 32 (8.9) | 35 (9.8) | 0.899 |
| Cycling | 31 (8.7) | 33 (9.2) | 0.944 | Power exercise | **31 (8.7)** | **55 (15.4)** | 0.375 |
| Swimming | 29 (8.1) | 16 (4.5) | 0.646 | Badminton | 30 (8.4) | 24 (6.7) | 0.815 |
| Power exercises | 26 (7.3) | 28 (7.8) | 0.945 | Swimming | 24 (6.4) | 21 (2.2) | 0.495 |
| Badminton | 24 (6.7) | 26 (7.3) | 0.934 | Skating | 23 (6.4) | 8 (5.9) | 0.960 |
| Volleyball | 23 (6.4) | 27 (7.6) | 0.869 | Track and field | 21 (5.9) | 23 (6.4) | 0.945 |
| Basketball | 19 (5.3) | 15 (4.2) | 0.882 | Basketball | 14 (3.9) | 9 (2.5) | 0.856 |
| Table tennis | 8 (2.2) | 6 (1.7) | 0.947 | Table tennis | 14 (3.9) | 13 (3.6) | 0.967 |
| Skating | 7 (2.0) | 4 (1.1) | 0.911 | Volleyball | 14 (3.9) | 15 (4.2) | 0.967 |
| Czech girls | | | | | | | |
| Cycling | 152 (17.2) | 128 (14.5) | 0.539 | Downhill skiing | 247 (28.0) | 213 (24.2) | 0.356 |
| Swimming | 150 (17.0) | 120 (13.6) | 0.443 | Track and field | 80 (9.1) | 87 (9.9) | 0.860 |
| Track and field | 149 (16.9) | 156 (17.7) | 0.854 | Snowboarding | 71 (8.0) | 89 (10.1) | 0.647 |
| Skating | 68 (7.7) | 61 (6.9) | 0.862 | Power exercises | 69 (7.8) | 80 (9.1) | 0.777 |
| Power exercises | 52 (5.9) | 41 (4.7) | 0.799 | Swimming | 68 (7.7) | 47 (5.3) | 0.613 |
| Walking | 47 (5.3) | 30 (3.4) | 0.697 | Skating | 62 (7.0) | 48 (5.4) | 0.732 |
| Volleyball | 47 (5.3) | 64 (7.3) | 0.672 | Dancing | 38 (4.3) | 49 (5.6) | 0.783 |
| Dancing | 38 (4.3) | 53 (6.0) | 0.721 | Walking | 32 (3.6) | 20 (2.3) | 0.792 |
| Badminton | 30 (3.4) | 18 (2.0) | 0.779 | Volleyball | 29 (3.3) | 31 (3.5) | 0.966 |
| Tennis (soft tennis) | 19 (2.2) | 18 (2.0) | 0.966 | Cross-country skiing | 27 (3.1) | 34 (3.9) | 0.867 |
| Polish girls | | | | | | | |
| Track and field | 55 (12.8) | 64 (14.9) | 0.742 | Skating | 58 (13.5) | 53 (12.3) | 0.851 |
| Swimming | 50 (11.6) | 41 (9.6) | 0.759 | Downhill skiing | 48 (11.2) | 39 (9.1) | 0.784 |
| Volleyball | 45 (10.5) | 46 (10.7) | 0.975 | Swimming | 40 (9.3) | 32 (7.4) | 0.773 |
| Cycling | 37 (8.6) | 38 (8.9) | 0.963 | Gymnastics | 35 (8.1) | 28 (6.5) | 0.809 |
| Skating | 37 (8.6) | 29 (6.8) | 0.787 | Snowboarding | 23 (5.3) | 23 (5.3) | 1.000 |
| Badminton | 32 (7.4) | 30 (7.0) | 0.952 | Power exercises | 23 (5.3) | 28 (6.5) | 0.857 |
| Power exercises | 21 (4.9) | 17 (4.0) | 0.894 | Badminton | 22 (5.1) | 20 (4.7) | 0.952 |
| Gymnastics | 18 (4.2) | 25 (5.8) | 0.815 | Track and field | 21 (4.9) | 37 (8.6) | 0.602 |
| Dancing | 17 (4.0) | 20 (4.7) | 0.917 | Volleyball | 19 (4.4) | 24 (5.6) | 0.859 |
| Aerobics | 16 (3.7) | 14 (3.3) | 0.953 | Aerobics | 15 (3.5) | 13 (3.0) | 0.941 |

PA: physical activity; *p*: significance level.

**Table 5.** Organized physical activity among LA (tertile 1) and HA (tertile 3) Czech and Polish boys and girls.

| | | | | Organized Physical Activity | | | | |
|---|---|---|---|---|---|---|---|---|
| **Type of PA** | **Low PA** | **High PA** | ***p*** | **Type of PA** | **Low PA** | **High PA** | ***p*** |
| | ***n* (%)** | ***n* (%)** | | | ***n* (%)** | ***n* (%)** | |
| Czech boys | | | | Polish boys | | | |
| Soccer | 85 (15.8) | 120 (22.3) | 0.248 | Soccer | 58 (16.2) | 92 (25.8) | 0.168 |
| Floorball | 36 (6.7) | 50 (9.3) | 0.665 | Volleyball | 25 (7.0) | 32 (9.0) | 0.784 |
| Track and field | 25 (4.6) | 30 (5.6) | 0.867 | Martial arts | 19 (5.3) | 27 (7.6) | 0.758 |
| Martial arts | 24 (4.5) | 27 (5.0) | 0.933 | Basketball | 15 (4.2) | 14 (3.9) | 0.967 |
| Power exercise | 22 (4.1) | 24 (4.5) | 0.947 | Swimming | 15 (4.2) | 19 (5.3) | 0.882 |
| Basketball | 16 (3.0) | 21 (3.9) | 0.883 | Gymnastics | 14 (3.9) | 11 (3.1) | 0.915 |
| Dancing | 15 (2.8) | 11 (2.0) | 0.897 | Track and field | 12 (3.4) | 17 (4.8) | 0.854 |
| Volleyball | 11 (2.0) | 12 (2.2) | 0.973 | Power exercise | 11 (3.1) | 22 (6.2) | 0.705 |
| Tennis | 9 (1.7) | 6 (1.1) | 0.924 | Badminton | 10 (2.8) | 11 (3.1) | 0.968 |
| Shooting | 7 (1.3) | 3 (0.6) | 0.922 | Tennis | 7 (2.0) | 3 (0.8) | 0.891 |
| Czech girls | | | | Polish girls | | | |
| Dancing | 91 (10.3) | 118 (13.4) | 0.495 | Volleyball | 52 (12.1) | 69 (16.0) | 0.544 |
| Track and field | 66 (7.5) | 65 (7.4) | 0.983 | Gymnastics | 30 (7.0) | 15 (3.5) | 0.637 |
| Volleyball | 65 (7.4) | 72 (8.2) | 0.862 | Dancing | 27 (6.3) | 38 (8.8) | 0.711 |
| Aerobics | 26 (2.9) | 38 (4.3) | 0.772 | Basketball | 20 (4.7) | 29 (6.7) | 0.770 |
| Basketball | 17 (1.9) | 27 (3.1) | 0.809 | Badminton | 12 (2.8) | 8 (1.9) | 0.898 |
| Martial arts | 17 (1.9) | 25 (2.8) | 0.853 | Soccer | 12 (2.8) | 7 (1.6) | 0.868 |
| Power exercises | 15 (1.7) | 47 (5.3) | 0.555 | Swimming | 11 (2.6) | 23 (5.3) | 0.720 |
| Horse riding | 15 (1.7) | 32 (3.6) | 0.722 | Track and field | 10 (2.3) | 26 (6.0) | 0.647 |
| Tennis | 15 (1.7) | 19 (2.2) | 0.917 | Skating | 9 (2.1) | 4 (0.9) | 0.878 |
| Swimming | 15 (1.7) | 23 (2.6) | 0.855 | Power exercises | 9 (2.1) | 9 (2.1) | 1.000 |

PA: physical activity; *p*: significance level.

**Table 6.** Odds ratios for meeting the 3 × 20 min VPA recommendation by participation in OPA and the most frequently practiced types of PA throughout the year.

| Variables | Low PA Boys—CZ | | Low PA Boys—PL | | Low PA Girls—CZ | | Low PA Girls—PL | |
|---|---|---|---|---|---|---|---|---|
| | **OR (95% CI)** | ***p*** | **OR (95% CI)** | ***p*** | **OR (95% CI)** | ***p*** | **OR (95% CI)** | ***p*** |
| Organized PA | 3.16 (1.63–6.15) | 0.001 | 2.10 (1.01–4.36) | 0.046 | 2.16 (1.21–3.86) | 0.006 | | |
| Cycling | | | | | | | 2.63 (1.17–5.94) | 0.020 |
| Power exercise | | | 3.40 (1.67–6.95) | 0.001 | 2.42 (1.29–4.54) | 0.006 | | |
| Badminton | | | 3.23 (1.43–7.29) | 0.005 | | | | |
| Soccer | 2.03 (1.02–4.05) | 0.044 | | | | | | |

OR, odds ratio; CI, confidence interval; PA, physical activity; *p*, significance level. Categorial covariates, age, badminton, BMI, basketball, cycling, dancing, downhill, floorball, organized physical activity, power exercise, skating, soccer, swimming, track and field, volleyball, and walking.

## 4. Discussion

There is a lack of current studies prioritizing associations between preferred and practiced types of PA among LA boys and girls. This study was the first to provide an overview of preferred and practiced types of PA among Czech and Polish LA boys and girls in comparison with HA boys and girls. It was confirmed that LA boys and girls preferred PA types that were less demanding in terms of movement and fitness, as well as health-related PA. When promoting these types of PA, it is important to bear in mind that the benefit of PA for health is insufficient as a motivation for the actual practice of PA [35]. Moreover, the feeling of pressure, being evaluated, and not receiving support, are demotivators in performing PA among LA adolescents [27].

The differences between LA and HA adolescents are affected by gender differences in PA, which have increased over the years [2]. These gender differences also affect

the preferred types of PA between boys and girls [36,37]. Nielsen et al. [38] stated that although Denmark has had a long tradition of gender-integrated PE, very traditional gender differences remained similar to countries with gender-segregated PE. Of Danish boys, 85% preferred ball games (only 59% of girls), while 62% preferred dance (only 16% of boys). According to a study by Metcalfe and Lindsey [39], boys prefer traditional team sports, whereas young women choose to engage in gym/fitness activities to promote appearance and feminine attractiveness. Just as gender differences are not sufficiently respected in integrated PE [38,40], they are not respected even in gender-segregated PE [41].

Resaland et al. [42] reported that gender differences are clearly observed among younger girls by their preference for dancing and exercising to music. The authors also suggested that children who were less fit could be offered activities such as frisbee, dodge-ball, and floorball. According to Peral-Suárez et al. [43], there is a lack of information on PA practice and sports preferences among children in terms of gender, and that this may increase gender inequalities. They found that even in childhood, girls preferred individual sports with artistic connotations, while boys often practiced more team contact sports. Interventions aimed at increasing girls' participation in team sports may only encourage girls to try team sports, but their impact on sustained participation and subsequent PA outcomes is less apparent [44]. Gender-neutral access to PA is particularly important for LA adolescents [39]. It should also be taken into account that high levels of depression in girls and high levels of aggression in boys seem to be relatively stable across time [45] and that gender differences will affect PA preferences.

To date, there have been few studies on PA preferences available. In recent years, more attention has been paid to researching PA preferences among children [42,46], adolescents [47,48], fitness PA [33,37], tourism activity [49], and PA preferences in comparison with sedentary behavior [50]. No studies are available on the preferences of adolescents with LA. The main issue associated with the lack of information is the difficulty of diagnosing PA preferences, its variability, regional dependence, changing sporting achievements of national team members, changing media coverage of sports, gender and age specifics, and other influences. Another serious aspect is the insufficient theoretical background for the assessment of preferences, inclinations, favoring, wishes, motivations, or interests, as well as the variability and reliability of research methods [46,51]. Even the category preference system applied in the present study does not have adequate taxonomic support in the classification of PA types into the selected categories according to the criteria of organization, prevailing PA environment, and prevailing PA focus.

An important finding of the study is that active participation in OPA among Czech and Polish LA boys and Czech LA girls increased their likelihood of meeting VPA recommendations. The insignificant effect of participation in the OPA of Polish girls on meeting VPA recommendations is not consistent with the results of a study by Groffik et al. [14]. The increase in the number of hours of active participation in OPA has a significant effect on both boys and girls in the achievement of at least $3 \times 20$ min of VPA. Despite these controversial findings, which require further research, the focus on increasing OPA among adolescents with low PA should be a priority both in school programs and for economically subsidized community PA programs, including changes in state and school policies. The role of schools is also crucial because low PA and non-participation in sports are correlated with lower socioeconomic status and low parental education [52]. Tassitano et al. [53] suggested that all youth should have access to a PA-promoting structured setting, which is not possible without state subsidy during the post-pandemic period.

In order to increase PA in LA adolescents, it is more important to use objective indicators of PA levels measured by wearable devices than in HA adolescents. Most adolescents believe themselves to be more physically active than they really are [54]. Non-awareness of potential health risks is a concern, especially for LA individuals. School interventions aimed at supporting PA in adolescents with low physical activity should focus more on motivation in areas of enjoyment, perceived autonomy, intrinsic motivation, motivational climate, and goal orientation [55]. Another important aspect is fostering

perceived motor competence in PE to improve motivation for PA [56]. According to Palmer-Keenan and Bair [57], the preferred types of PA by LA adolescents should be (1) enjoyable (e.g., dancing, with friends and family); (2) comfortable (indoors, not sweaty, and not physically competitive or embarrassing); and (3) promoted by "cool" and reliable personnel (e.g., teens such as themselves or young comedians).

In order to achieve positive changes resulting in physically active behaviors among LA adolescents, further research studies should investigate the effects of reducing sedentary time rather than promoting PA. Interventions aimed at children that focused on sedentary behaviors have resulted in greater effects in daily PA [54].

In sum, school-based support for PA aimed at low physically active individuals should focus on the following:

- Less demanding types of PA in terms of movement and fitness and less "competitive" types of PA;
- Socially attractive types of PA with less contact;
- Types of PA that are strongly health related;
- Widely applicable types of leisure PA that can be pursued individually and in groups;
- Lifelong PA that are less time consuming and financially affordable;
- PA associated with the needs of everyday life, especially transport.

This study is the first to present an overview of the types of PA preferred and practiced by LA versus HA adolescents. The findings highlight new perspectives that may be relevant to addressing persistent gender differences in PE and PA among adolescents in secondary schools, as well as some methodological issues in the diagnosis of PA preferences.

A limitation of the study is the category system of evaluating physical activity preferences based on three criteria that do not allow a clear classification of PA types. However, the category system was implemented because it is more credible than the more demanding approach of choosing from dozens of PA types.

## 5. Conclusions

In addition to presenting gender differences in the preference for PA types, this study confirmed that there is a difference in the preference and practice of PA types between high and low active boys and girls. Physically low active boys and girls preferred PA types that were less demanding in terms of movement and fitness, as well as health-oriented PA. The practice of different PA types was not so significant, which suggests that there may be an insufficient emphasis on the preferred types of PA and the preferred PA that focus on providing opportunities for PA participation. Increasing active participation of adolescents who are physically low active in OPA should be a priority objective for secondary schools and out-of-school institutions. Ensuring "equal" access for all adolescents is essential, especially after the pandemic. The global PA recommendation and the monitoring of PA in children and youth should include continuous assessment and analyses of trends in the preferences and practices of different PA types, especially those who are less physically active.

**Author Contributions:** K.F. conceptualized the study design and contributed to the analyses and writing of the manuscript. D.G. contributed to data collection and writing of the manuscript. M.K. analyzed the data and project administration. M.Š., writing—revision and editing. A.Z., writing—revision and editing. J.M., supervision, revision, and editing. All authors have read and agreed to the published version of the manuscript.

**Funding:** This research was funded by the research grant of Czech Science Foundation (No. 13-32935S): The objectification of comprehensive monitoring of school mental and physical strain in adolescents in the context of physical and mental condition.

**Institutional Review Board Statement:** The study was conducted according to the guidelines of the Declaration of Helsinki and approved by the Ethics Committee of the Faculty of Physical Culture, Palacký University Olomouc (No. MSM6198959221 and No. 37/2013).

**Informed Consent Statement:** Informed consent was obtained from all subjects involved in the study.

**Data Availability Statement:** The data presented in this study are available upon request from the corresponding author.

**Conflicts of Interest:** The authors declare no conflict of interest.

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
