# Peer review of "The Differences in Physical Activity Preferences and Practices among High versus Low Active Adolescents in Secondary Schools"

_sustainability, doi:10.3390/su14020891_

Round 1
Reviewer 1 Report
Thank you for the opportunity to review this manuscript. It is interesting to see the differences of interests about the different sports depending on the level of PA practiced.
There are some aspects to improve:
Using more homogeneous language throughout the manuscript, there are times when we speak of adolescents and other times of boys and girls in many cases without the need to see the difference by sex.
The study has been done in secondary schools, but other times it only talks about schools.
In Table 1, na "n" is always lowercase.
There are no results for the three groups based on tertiles on the PA level. It would be interesting to reflect them in the manuscript.
Author Response
There are some aspects to improve:
Using more homogeneous language throughout the manuscript, there are times when we speak of adolescents and other times of boys and girls in many cases without the need to see the difference by sex.
Response: Dear Reviewer, we would like to thank you for the opportunity to review our manuscript. We have adjusted the terminology as much as possible to fit the need to see the differences by sex, the rest we left in uniform use.
The study has been done in secondary schools, but other times it only talks about schools.
Response: We have revised carefully this misleading terminology.
In Table 1, na "n" is always lowercase. We adjusted.
There are no results for the three groups based on tertiles on the PA level. It would be interesting to reflect them in the manuscript.
Response: Thank you for the comment. We have the results in three tertiles processed in the tables, but their inclusion reduces the clarity of the tables. The addition of all tables would significantly expand the scope of the study with no additional interesting findings therefore we compare only the lowest and the highest tertiles.
Reviewer 2 Report
Comments
Abstract and Introduction
Comment 1: The Abstract specified that study variables included preferred, practiced, and organized PA. Please add the tool measuring the organized PA which was omitted in “the preferences and practices of the different types of PA were identified using a PA preference questionnaire and weekly PA was identified using the IPAQ-long form”.
Comment 2: In the fourth paragraph of the Introduction, the role of school in students’ health promotion is highlighted. It seems that the authors would measure PA organized by the school. But in the measure section, it seems that the organized PA was measured by the question “Indicate your participation in regular and organized PA (under supervision of a teacher, trainer, or coach) during the week in your free time during the past 12 months, except for holidays. Was this question particularly asking the organized PA in schools?
Comment 3: The research aims are needed to be more clarified. Did the organized PA refer to PA organized by schools or any PA in an organized form (e.g., sports clubs outside schools)?
Methods
Comment 4: Table 1 should be outlined in the Results. Line 90-137: Font size should be consistent with other content. Line 97 and line 138: please remove duplicates.
Comment 5: Line 90: What does “similar conditions” mean? please add more details.
Comment 6: Please add the information about the reliability and validity of the questionnaire on preference for physical activity (QPAP).
Results
Comment 7: In Line 201, the number of subtitle “Differences in the practice of PA types” should be “3.3”. And in Line 224, there is a repetition number of “3.5” of subtitle “Achievement of vigorous physical activity recommendations”.
Comment 8: The numbers can be found in the table. Please try not to repeat them in the main text again unless they are especially important. Please make the results more concise, and this comment also applies to the rest of the result section.
Discussion
Comment 9: When discussing the gender difference, please make more comparison between your main findings and results from previous results, and explain possible reasons in detail.
Comment 10: The introduction did not mention whether some previous studies have explored this topic. The authors need to clearly specify what is the novelty of your research compared with current published work. Finally, the author should compare and link the combined results in the discussion section with previous related studies, as to make this article more valuable.
Conclusion
Comment 11: The conclusion should be compendious.
Author Response
Comments
Abstract and Introduction
Comment 1: The Abstract specified that study variables included preferred, practiced, and organized PA. Please add the tool measuring the organized PA which was omitted in “the preferences and practices of the different types of PA were identified using a PA preference questionnaire and weekly PA was identified using the IPAQ-long form”.
Response: Dear Reviewer, we would like to thank you for the opportunity to review our manuscript. We added. The preferences and practices of the different types of PA and participation in organized PA were identified using a PA preferences questionnaire,
Comment 2: In the fourth paragraph of the Introduction, the role of school in students’ health promotion is highlighted. It seems that the authors would measure PA organized by the school. But in the measure section, it seems that the organized PA was measured by the question “Indicate your participation in regular and organized PA (under supervision of a teacher, trainer, or coach) during the week in your free time during the past 12 months, except for holidays. Was this question particularly asking the organized PA in schools?
Response: The question covers all types of any organized PA in the free time and did not include PA in schools including PE lessons. The distinction between PE lessons, extracurricular organized forms of PA in school and other types of organized leisure PA requires deeper diagnostics, which is further complicated by the specifics of Czech and Polish schools. Unfortunately, this interesting issue could not be addressed in the long term in the research of both countries and in this study. In the text, we emphasized that organized PA included any PA in an organized form except of PE lessons.
Comment 3: The research aims are needed to be more clarified. Did the organized PA refer to PA organized by schools or any PA in an organized form (e.g., sports clubs outside schools)?
Response: We specified this in the methods. Regular organized PA included any PA in an organized form except of PE lessons.
Methods
Comment 4: Table 1 should be outlined in the Results.
Response: We moved Table 1 to the results.
Line 90-137: Font size should be consistent with other content. Line 97 and line 138: please remove duplicates.
Response: Duplicates removed and font size uniformed.
Comment 5: Line 90: What does “similar conditions” mean? please add more details.
Response: We added “at the time of the habitual weather (same season) and the habitual educational weekly program.”
Comment 6: Please add the information about the reliability and validity of the questionnaire on preference for physical activity (QPAP).
Response: We added. The highest stability between the first and the second measurements were in the group of team sports (rs = 0.76 – 0.81) and the lowest stability was then recorded for the group of rhythmic and dancing activities (rs = 0.62 – 0.68).
Results
Comment 7: In Line 201, the number of subtitle “Differences in the practice of PA types” should be “3.3”. And in Line 224, there is a repetition number of “3.5” of subtitle “Achievement of vigorous physical activity recommendations”.
Response: We have revised and corrected.
Comment 8: The numbers can be found in the table. Please try not to repeat them in the main text again unless they are especially important. Please make the results more concise, and this comment also applies to the rest of the result section.
Response: We have reduced the interpretation of the results.
Discussion
Comment 9: When discussing the gender difference, please make more comparison between your main findings and results from previous results, and explain possible reasons in detail.
Response: We are aware of this shortcoming. Unfortunately, the issue of preferences is theoretically underdeveloped, and little research addresses this serious issue. Therefore, we often have to compare the results with the previous results of our research, which is not entirely ideal for comparison in the discussion. Comparison also makes it difficult to understand different preferences, inclinations, interests, relationships, popularity, etc.
Comment 10: The introduction did not mention whether some previous studies have explored this topic. The authors need to clearly specify what is the novelty of your research compared with current published work. Finally, the author should compare and link the combined results in the discussion section with previous related studies, as to make this article more valuable.
Response: We added to the discussion: There is a lack of current studies prioritizing associations between preferred and practiced types of PA among LA boys and girls. This study was the first to provide such comparison and on international level.
Conclusion
Comment 11: The conclusion should be compendious.
Response: We made minor adjustments.
Round 2
Reviewer 1 Report
The authors have improved the suggestions made.